# Insight into Cancer Immunity: MHCs, Immune Cells and Commensal Microbiota

**DOI:** 10.3390/cells12141882

**Published:** 2023-07-18

**Authors:** Minting Wen, Yingjing Li, Xiaonan Qin, Bing Qin, Qiong Wang

**Affiliations:** School of Life Science, Guangzhou University, Guangzhou 510006, China

**Keywords:** MHC-I, tsMHC-II, NK cells, γδ T cells, eosinophils, microbiota, cancer immunity

## Abstract

Cancer cells circumvent immune surveillance via diverse strategies. In accordance, a large number of complex studies of the immune system focusing on tumor cell recognition have revealed new insights and strategies developed, largely through major histocompatibility complexes (MHCs). As one of them, tumor-specific MHC-II expression (tsMHC-II) can facilitate immune surveillance to detect tumor antigens, and thereby has been used in immunotherapy, including superior cancer prognosis, clinical sensitivity to immune checkpoint inhibition (ICI) therapy and tumor-bearing rejection in mice. NK cells play a unique role in enhancing innate immune responses, accounting for part of the response including immunosurveillance and immunoregulation. NK cells are also capable of initiating the response of the adaptive immune system to cancer immunotherapy independent of cytotoxic T cells, clearly demonstrating a link between NK cell function and the efficacy of cancer immunotherapies. Eosinophils were shown to feature pleiotropic activities against a variety of solid tumor types, including direct interactions with tumor cells, and accessorily affect immunotherapeutic response through intricating cross-talk with lymphocytes. Additionally, microbial sequencing and reconstitution revealed that commensal microbiota might be involved in the modulation of cancer progression, including positive and negative regulatory bacteria. They may play functional roles in not only mucosal modulation, but also systemic immune responses. Here, we present a panorama of the cancer immune network mediated by MHCI/II molecules, immune cells and commensal microbiota and a discussion of prospective relevant intervening mechanisms involved in cancer immunotherapies.

## 1. Introduction

In recent decades, cancer immunotherapies like chimeric antigen receptor-engineered T cell (CAR-T), checkpoint blockade of PD-1/PD-L1, etc., are emerging in an endless stream which arrests the attention of physicians, patients and researchers from all over the world for their different, genius and extraordinary efficiency on cancer treatment compared with traditional chemotherapy and radiotherapy. This demonstrates that cancer generation and development are closely bound up with immunological events in the body.

In general, extrinsic pathogens and abnormal intrinsic cells with senescent, malignant transforming, infected, dysfunctional or other phenotypes should be targeted by immune cells and, in consequence, be killed. In specific immune responses to the special antigens, there are major histocompatibility complexes, commonly including MHC-I and MHC-II molecules that play key roles in cell-intrinsic and pathogen-extrinsic antigen targeting and presentation, respectively. All nucleated cells in the body express MHC-I molecule and cellular-intrinsic antigens are thereby processed and presented by MHC-I. For example, an infected cell combines pathogen peptide with MHC-I and presents the complex, or a malignant cell combines cancerous peptide with MHC-I and presents the complex. Those complexes can be recognized by cytotoxic T lymphocytes (CTLs), such as CD8^+^T cells, and apoptosis of the targeted cell is initiated as a consequence, while extrinsic pathogens, including bacteria, fungi, viruses, chlamydia or mycoplasma, can be recognized and engulfed by antigen-presenting cells (APCs) via endocytosis. Thereafter, extrinsic peptides are presented by APCs through being combined with MHC-II molecules. Usually, MHC-II molecules are identified by CD4^+^T cells which then play a role as a helper in mediating CD8^+^T cell activation and inducing B lymphocyte differentiation, and then the latter two initiate cellular immunity and humoral immunity, respectively.

## 2. MHC-I Reduction and Immune Evasion

In cancer immune surveillance, T lymphocyte-mediated specific immune response plays a crucial role. However, cancer immunity can be blocked through the alteration of the tumor microenvironment, leading to surveillance evasion. Thus far, the related strategies include: decreasing cancer antigens’ immunogenicity and rendering cancer cells less visible, muting APCs, inhibiting naïve T cell stimulation, depressing CTL infiltration into the tumor, suppressing CTLs’ cytotoxic efficacy, etc. [1,2,3]. Clinical cancer immunotherapies may, therefore, suffer from cancer cell-mediated immune invisibility through immune evasion mechanisms, which usually result from the downregulation of MHC-I expression or the failure of antigen presentation (Figure 1). In human cells, MHC-I molecules are known as human leukocyte antigens (HLA) and HLA class-I molecules are composed of classical (class-Ia) and the non-classical (class-Ib) components [4,5,6,7]. In fact, HLA-Ia expression defects are always found followed by uncontrolled growth of both primary and metastatic cancers [8,9,10], demonstrating their correlation with high tumor grading and progression, decreased survival and ineffective immunotherapies [11,12,13,14]. An MHC-I expression defect could derive from reversible lesions and irreversible lesions. The former are aroused from epigenetic modifications such as IFNγ-induced MHC-I expression [15,16], while the latter are aroused from less common structural mutations. Through investigating the regulation of MHC-I expression, Kobayashi et al. found that NOD-, LRR- and CARD-containing 5 (NLRC5) notably upregulated MHC-I, β2-microglobulin (β2M) and transporter associated with antigen processing 1 (TAP1), while the latter two as indispensable parts play key roles in the MHC-I antigen presentation pathway [17]. This study also revealed that NLRC5 was induced by IFNγ in a dose-dependent manner, indicating that NLRC5 probably plays a key role in the MHC-I antigen presentation pathway. Subsequently, they also reported in the following studies that NLRC5 expression was lost in varying cancers and the loss was correlated with the loss of CTL activation in several types of cancer [18].

It seems that NLRC5 expression is not invariable in different tissues according to the studies: high levels were shown in almost hematopoietic cells and tissues, particularly in CD8^+^ and CD4^+^ T lymphocytes, B lymphocytes, Natural Killer (NK) cells and NK-T [19,20,21,22,23], whereas intermediate levels were shown in macrophages and dendritic cells [21,24,25,26]. Upregulated MHC-I expression is performed not only through the NLRC5 pathway itself, but also in a synergistic way with the help of additional cis-regulatory elements’ (ISRE and enhancer A) activation. Their expression is induced by IFNγ-caused STAT1 homodimer formation, which activates IRF1 and NLRC5 gene promoters through binding gamma-activated sequence (GAS) [27]. This was also confirmed by the involvement of activated IFN and STAT1 in the increase in NLRC5 expression during the activation of CTLs and T helper (Th) cells [21]. In vitro, the efficiency of TCR transgenic CD8^+^ T cells was activated and supported by NLRC5-deficient bone marrow-derived macrophages (BMDM) provided with SINFEKKL peptide, which was as similar as the efficacy of CD8^+^ T cells induced by wild-type BMDCs, while it was failed with the application of NLRC5-deficient cells provided with negative control [28]. After in vivo intravenous loading of Listeria monocytogenes, the IFN-producing CD8^+^ T cell was not aroused to an increase in the cell number in the spleen or the liver of NLRC5-deficient mice, consequently resulting in severe infection compared with the CD8^+^ T cell in wild mice [28,29]. In addition to CD8^+^ and CD4^+^ T lymphocytes, NLRC5 is implicated in γδT cell activation as well and its expression positively correlated with the induction of butyrophilin (BTN) family protein BTN3A1-3, whose gene contains SXY modules W/S, X1, X2 and Y boxes as cis-regulatory elements of the MHC-II gene promoter [30]. All these studies suggest that NLRC5 plays a role in the activation of MHC-I transcription, and the facilitation of CD8^+^ and CD4^+^ T lymphocyte development and immune function initiation (killing or helper).

In spite of numbers of studies proving NLRC5 expression in different tissues and its role in the MHC-I antigen presentation pathway [19,20,21,22,23], the regulatory mechanisms of the expression have not been well-elicited. These studies suggest that Chromatin NLRC 5 promoter accessibility is varied and modulated by different transcription factors in different cell types, where epigenetic regulation of chromatin remodeling may be a key pathway potentially adjusting NLRC 5 expression, such as in human cancers as well as cancer cell lines. NLRC 5 was involved in MHC-I expression attenuation through regulating promoter hypermethylation [18]. NLRC 5 expression, like other functional proteins, can also be regulated at the post-transcriptional and post-translational levels by long non-coding RNAs (lncRNAs) [31,32,33,34,35,36]. Interestingly, it was recently reported that MHC-I and APM expression was reduced by depressing NLRC5 in quiescent hair follicle stem cells and muscle stem cells in order to protect from immune surveillance and destruction [37]. Additionally, the reduction of MHC-I in cancer-originating cells of different types of cancer disrupted CTL-mediated damage, leading to the formation of variants that could escape immune surveillance and thereafter causing cancer recurrence [38,39,40,41,42,43]. Compared with immune evasion, these may suggest that NLRC5 and MHC-I reduction may facilitate the formation of a global dedifferentiation phenotype in cancer cells, which suggests the progression of cancer growth and the initiation of cancer cell communities.

As NLRC5 plays key role in the activation of MHC-I expression which is commonly lost in cancer immune evasion, NLRC5 could be the point that can switch on antitumor immunity. Several relevant in vitro and in vivo studies suggested that MHC-I expression in tumor cells could be more necessary than that expression in APCs to induce protective antitumor immunity [44,45]. Therefore, NLRC5 expression acquisition in tumor cells can enhance their MHC-I expression and antigen presentation and put forward antitumor immune response and CTL-mediated destruction [46,47,48]. However, in contradiction with the studies above focusing on the antitumor aspects of NLRC5, there are also studies implicating the role of NLRC5 in promoting tumor growth. According to their reports, NLRC5 was shown to participate in vitro in modulating the biological behaviors of HCC, ccRCC, glioma and ESCC cells, such as promoting cancer cell growth, motility and migration, and played a similar role in mice lacking the functionality of an adaptive immune system [33,36,49,50,51,52,53], while NLRC5 in mice with a competent immune system evidently suppressed the growth of melanoma and pancreatic ductal adenocarcinoma (PDAC) cells [54,55]. Considering that NLRC5 and MHC-I could be involved in progressive cancer growth and cancer cell population initiation, those contrary phenomena may result from that NLRC5-elicited antitumor immune response overcoming NLRC5-stimulated cancer growth in cancer cells originating from mice with normal immune function.

Studies on how to compensate for insufficient MHC-I expression and blocked antigen presentation in tumor cells are still progressing. IFN has been shown to potentially reverse the defect in MHC-I expression and accelerate antigen processing in cancer cells through the modification of proteasome constituents [45]. However, the limitation of potential use of IFN is there, where its systemic toxicity and development of cancer unresponsiveness to IFN are involved [56,57,58]. Epigenetic drugs may derepress NLRC5 showing the capacity to restore MHC-I expression, while their efficacy on solid tumors is limited and off-target effect is also a key challenge [59,60,61]. In general, genetic and epigenetic alteration-caused loss of NLRC5 paves the way for MHC-I deficiency, which is a common mechanism of cancer unresponsiveness to immunotherapy.

## 3. TsMHC-II Function and Tumor Immune Surveillance

Whereas MHC-Is are distributed primarily on the surface of almost all nucleated cells and mainly provide endogenous antigen to CD8^+^ T cells, MHC-IIs of APCs, such as dendritic cells (DCs), B cells and macrophages, primarily present exogenous antigens to CD4^+^ T cells. Though MHC-I-restricted neoantigens have shown to be important effectors for tumor-specific CTLs in the success of immune checkpoint inhibition (ICI) [62,63,64,65,66,67,68,69], prognostic indicators are not precise enough [70,71,72,73,74]. Therefore, considering CD4^+^ T cells play important auxiliary roles in the activation of CD8^+^ T cells and the formation of memory T cells [75,76,77,78,79,80], the influence of MHC-II-restricted CD4^+^ T cells has been addressed and recognized as being effective [81,82]. In fact, besides APCs, other cell types including tumor cells are able to express MHC-II as well [71]. Tumor-specific MHC-II expression (tsMHC-II) can facilitate immune surveillance to detect tumor antigens (Figure 2), and thereby has been used in immunotherapy, including superior cancer prognosis, clinical sensitivity to ICI therapy and tumor-bearing rejection in mice [72,73,74,75,76,77,81,82]. Clinically, tsMHC-II has been seen in a variety of human tumors including: breast cancer [73,74,83,84,85,86,87,88,89,90], colorectal cancer [91,92], classic Hodgkin lymphoma [76], glioma [93], melanoma [72,77,94], non-small cell lung cancer [95], ovarian cancer [96,97] and prostate cancer [98].

In the mechanism, those functions of tsMHC-II may be derived from the stimulation of CD4^+^ T cell subsets, of which Th1 CD4^+^ cells express IFN-γ and other cytokines, while regulatory T cells (Tregs) play a suppressive role in immunity and inflammation and contribute to tolerance [99]. For other CD4^+^ T cell subsets like Th2 and Th17, the mechanisms are still ambiguous for their both pro-tumor and anti-tumor efficacy in cancer immunity [99]. Usually, antigens can be presented by MHC-II from a greater repertoire. This causes CD4^+^ T cells to be capable of recognizing cancer-associated antigens in a wider array than CD8^+^ T cells [100,101,102]. As tumors have fewer candidate neoantigens compared with normal cells [80], this specially indicates the functional significance of CD4^+^ T cells for tumor treatment. However, the exact effect of tsMHC-II on CD4+ Tregs in tumors is still not clear. In Tregs, TCR stimulation is performed through MHC-II as well to maintain suppressive activity [103], while in tumors in view of Tregs abundance [104,105] and tsMHC-II effect of improving immune-mediated outcomes, it could be suggested that tsMHC-II may be unable to stimulate TCR in Tregs. Normally, signal 1 (MHC:T cell receptor binding) and signal 2 (co-stimulation with CD80/86 binding to CD28) are required for T cell activation [106], while CD80/86 is not typically expressed by tumor cells [107]. Instead, non-classical co-stimulatory receptors beyond CD80/86 can be expressed by some cancer cells and are sufficient to activate certain T cell subsets [71,107], for example CD70, OX40-ligand, as well as molecules from adjacent, juxtaposed immune cells in the tumor microenvironment [108,109,110].

The expression of MHC-II as well as antigen presentation machinery is triggered by the transcriptional master regulator Class II Transactivator (CIITA). Without directly binding DNA, CIITA recruits transcriptional factors at promoter regions of MHC-II-related loci through scaffolding activity and induces the antigen presentation pathway where CIITA is necessary and sufficient [75,111]. CIITA plays a particularly important role in the process of IFNγ-initiated MHC-II expression, while some cancer cells appear insensitive to IFNγ stimulation [72,112]. Instead, multiple other pathways may be involved in regulating MHC-II expression in those cells. For example, JAK/STAT signaling is necessary for the upregulation of MHC-II expression [113,114], and MHC-II was suppressed by RAS/MAPK activation in breast cancer [83]. MAPK stimulated the CIITA promoter pIII in the HLA-DR^+^ melanoma cell line, and this promoter initiates transcription primarily in B cells [115]. Meanwhile, tumor cells may also downregulate MHC-II expression through some mechanisms. In Hodgkin Lymphoma, CIITA expression alteration was suffered in genomes, while tsMHC-II expression was blocked in primary large B-cell lymphoma [116,117,118]. Other non-genomic mechanisms are also involved. L-myc, N-myc and human achaete-scute complex homologue-1 were overexpressed and then interrupted the transcriptional response to IFNγ through binding to promoter IV of CIITA in small-cell lung cancer and neuroblastoma [95,119]. In addition, epigenetic alteration is a potential mechanism as well being involved in the reduction in MHC-II expression in human tumors [120]. This could link to cancer immune evasion since tumor cells can avoid recognition by anti-tumor CD4^+^ T cell subsets through inhibiting MHC-II expression. Therefore, repression of tsMHC-II expression may result in the resistance to ICI, and ensuring the correlation of the level of tsMHC-II expression with sensitivity to immunotherapy should be key to treatment and prognosis.

In fact, there could be a consequent correlation between tsMHC-II upregulation and tumor immunogenicity increase, suggested via the studies in mice. In clinic, tsMHC-II usually indicates improved prognosis, upregulation of favorable response to ICI, enlargement of TILs and pro-inflammatory IFN signaling in human tumors. This renders tsMHC-II a clinically operational biomarker reflecting sensitivity to the ICI, and strategies designed to increase the expression of the tsMHC-II could increase sensitivity to immunotherapy. Moreover, studies via murine models also infer that tsMHC-II should be a potential tool for anti-tumor vaccines. Mice were protected from injection with parental tumor cells through previous inoculation with non-viable tumor cells that express MHC-II [121,122,123,124]. Since studies showed that tsMHC-II upregulation often brings about tumor rejection, we may have a hypothesis that tsMHC-II expression promotion in human cancers could facilitate anti-cancer immunity and enhance sensitivity to ICI. Developing strategies to upregulate the levels of tsMHC-II in tumors through treatment, and gaining insights into its biological function, may bring new insights in cancer immunotherapy, but this requires more rigorous testing.

## 4. NK Cells and Anti-Cancer Immunity

While the priming of MHC-I expression and neoantigen presentation is known to introduce tumor antigen recognition performed by CTLs and subsequently trigger potent antitumor response, there are a number of strong scientific findings showing that immune responses elicited by non-CTLs like Natural Killer (NK) cells also play key roles in the regulation of cancer immunity. NK cells are a small subset of lymphocytes that account for innate immune response, in part including immunosurveillance and immunoregulation. NK cells are the early and rapid responders of the innate immune system and are inherently equipped with the capacity to recognize and capture pathogen-infected, damaged or transformed cells independent of the prophase sensitization process or antigen presentation. This distinct surveillance of NK cells is performed through the signaling that is triggered from surface-activating receptors or inhibitory receptors and most of the receptors can recognize molecules in a large repertoire including MHC-I, HLAs, chain-related proteins A and B molecules, nectin family proteins and others. Usually, the cytotoxic response of NK cell depends on the balance between the activating and inhibitory signals they receive (Figure 3). Responses are triggered under the condition of excessive activating signals [125]. NK cells also express FcgRIII receptors, which can recognize antibodies that are specific to tumor antigens and thereby perform antibody-dependent cell-mediated cytotoxicity (ADCC).

Compared with T cells that are targeting in an antigen-specific manner, NK cells, without targeting unique antigens, recognize expression patterns indicative of malignant transformation, that is to say, the pattern of the activating signals, such as the abnormal expression pattern of the molecules on the surface of a tumor cell could be preferentially recognized compared with the normal pattern of a healthy cell. The board of NK cells’ recognition is not specific, but broad, without prior training and independent on one unique molecule. Cytotoxic response triggered by NK cells could not be attenuated in case of the deficiency of unique antigens. Therefore, with similar functions and effects as CTLs, NK cells are capable of eliciting immune response to drug-resistant cancer cell populations. In contrast to CD8^+^ T cells, MHC deficiency annuls inhibiting signals towards tumor cells in NK cells and renders them more activating signals towards tumors. With this capability, NK cells have been shown to have exclusive capabilities to target highly malignant cancer stem-like cells and tumors with poorly differentiated or undifferentiated histological stages that are highly resistant to chemotherapy. Secondly, NK cells can catalyze differentiation of tumor cells through IFNγ secretion and stimulation [126], which causes remodeling of the surface receptor pattern of tumor cells, including an upregulation in MHC-I and CD54 and a downregulation in CD44 expression, followed by the disturbance of tumor growth and metastasis [127] and being well-recognized and eliminated by T cells. Thirdly, NK cells can selectively target aging tumor cells. Ruscetti et al. observed that KRAS-mutant lung tumor growth in mice was reduced after treatment with a cytostatic drug regimen with the effect that could primarily lead to natural aging of the body [128]. The evidence above supports the involvement of NK cells in checkpoint blockade and their impact on the response to immunotherapy.

In the immune system, NK cells participate in the initiation of the adaptive immune response by recruiting other immune cells, like DCs. In fact, through NK–DC crosstalk, NK cells release IFNγ and TNFα that promote DCs’ maturation [129], while DCs can facilitate NK cell proliferation, survival and activation through THE secretion of cytokines such as IL-12 and IL-15 [129,130]. Moreover, DCs, in case of misleading to maturation, will be killed in DC editing, which is performed by NK cells through the engagement of the activating receptor NKp30 [131]. Therefore, this crosstalk is of crucial significance in activating innate immune responses along with immune response to tumors. In the latter, DC–NK crosstalk has been considered an indicator of cancer immunotherapy after years of investigation [132,133]. Conventional type 1 DCs (cDC1) are important players in regulating cross-priming activation of T cells located in tumor-draining lymph nodes through secreted chemokines and cytokines that are involved in the modulation of T cell viability, capability and infiltration into the tumor microenvironment, which presents the importance of their clinical application for cancer immunity through highlighting their effect on tumor rejection abolishment, responsiveness to CAR-T therapy and ICI in mice with cDC1 deficiency [134,135]. For example, a recent study reported that induction and activation of cDC1s that reside within the tumors overcame acquired resistance and facilitated the tumors’ responsiveness to anti-PD-L1 therapy [136]. Actually, after activation, NK cells produce cDC1 chemoattractants and secrete them into the tumor microenvironment to attract cDC1s, and thereafter cDC1s can recruit CTLs and, in turn, elicit an adaptive immune response [137]. In clinic, increased populations of both tumor-infiltrating NK cells and cDC1 cells in the tumor microenvironment have been used as a biomarker for predicting and evaluating tumor responsiveness to PD-1 immunotherapeutic agents and prognosis of melanoma [138].

Oncolytic viruses (OVs), as a novel sort of anti-cancer drug, were first approved in 2015 and were followed rapidly by numerous other clinical trials. Originally, the therapeutic effect of Ovs was regarded to be due to viral completing replication within tumor cells, which in turn led to cell lysis. Now, it is recognized that OVs largely contributing to the induction of adaptive and innate immune responses could be responsible for the OV agents’ efficacy that has been reported [139]. That is to say, OVs can reduce the immunosuppressive pressure in the tumor microenvironment by priming the immune system. In clinic, NK cells are potentially involved in Ovs’ therapeutic efficacy via antiviral response of the innate immune system. Usually, the dual role of NK cells in innate defense leads to the removal of malignant cells and virus-infected cells, which meanwhile results in the clearance of the virus through killing virus-infected cancer cells and, thus, limits Ovs’ efficacy. Therefore, promoting OV-mediated lysis of tumor cells through endogenous NK cell depletion, together with adjuvant exogenous NK cell injection that augments the capacity of the immune system, could improve the efficacy of immunotherapy [140]. This was confirmed in a primary glioma mouse that was granted a significantly longer survival after being treated with the combination of OV therapeutics with activated human NK cells [141]. A better understanding and arrangement of the therapeutic variables such as Ovs’ nature, tumor stage, histological grade, dose regimen and adverse effects of the adoptive NK cell therapy could be necessary to optimize therapeutic interventions and improve the efficacy of antitumor immunity.

The efficacy of cancer immunotherapy has been clearly demonstrated to associate with the function of NK cells. Moreover, the incidence of cancers or other diseases could also be increased in case of NK cell dysfunction. In a prospective cohort study composed of healthy individuals, the capacity of immune response and disease incidence were investigated, showing that participants with low CTL cytotoxic activity had a higher risk for cancer incidence compared with those with medium or high cytotoxic activity [142]. Clinically, NK cells are dysfunctional or occur less frequently in most cancer patients and are further affected by surgical treatment and standard procedure of chemotherapy [143,144]. For example, fructose-bisphosphatase 1 could be inhibited by glycolysis, which leads to the dysfunction of NK cells and further tumor progression in KRAS-driven models of lung cancer [145]. Moreover, altered patterns of protein expression on the surface of tumor cells could also be the cause of dysfunctional NK cell response, such as radiation-caused increased PD-L1 level and decreased ligand level for Natural Killer Group 2D (NKG2D) receptor-protected non-small cell lung cancer cells from NK cell cytotoxicity [145,146,147,148]. Thus, surgery or treatment-caused NK cell dysfunction could pave the way to limited response or compromised efficacy to immunotherapies or related therapies targeting immune cells or molecules. In the adoptive NK cell therapy, endogenous NK cells were provided with a critical barrier in the setting of an altered tumor environment to attenuate their sensitivity to tumor immunosuppression. In addition, these cells were highly cytotoxic and targeting was ameliorated by chimeric antigen receptors to effectively increase the rates and duration of response [149,150].

## 5. γδ. T Cells and Anti-Cancer Immunity

According to the surface expression of αβ and γδ TCRs, T cells are primarily classified into two subsets: αβ T cells and γδ T cells. Whereas αβ T cells elicit an adaptive immune response through the recognition of target antigens that are presented via MHC molecules, γδ T cells account for 1–5% of total T cells compared with them, and are involved in the innate immune response that functions in a manner not by the MHC-introduced antigen presentation pathway but through the recognition of the surface proteins of the targeted cells, such as BTN and CD1 molecules (Figure 4) [151]. In addition, γδ T cells can bring us a new opportunity for “off-the-shelf” cellular therapeutics in allogeneic adoptive transfer settings since there is a low risk of the incidence of graft-versus-host disease with the application of γδ T cells clinically [152,153]. Currently, αβ T cells have been the focus of the majority of studies on T cell immunology and clinical application, whereas γδ T cells have also shown their obvious involvement in cancer immune processes and their potent killing effects on various types of cancer as reported by past studies [154,155,156,157]. For example, by analyzing the correlation between 22 different leukocyte subsets and cancer survival in 5782 clinical tumor samples, Gentles et al. proposed that tumor-residing γδ T cells might be the best predictor of overall patient survival [158].

The definition of human γδ T cell subtypes usually depends on the δ chain of γδ TCR, and the most relevant isoforms are, accordingly, Vδ1-3, where the most comprehensively reported isoforms are Vδ1 T cells and Vδ2 T cells, which both arise during the period of fetal development [159]. In adults, Vδ1 T cells maintain a population of about 10–15% of γδ T cells, primarily residing in peripheral tissues such as the gut epithelia, dermis, spleen and liver [160,161]. Vδ2 cells make up 80–85% of the total peripheral γδ T cells and are progressively predominant in the blood and spleen in adulthood [160]. The remaining γδ T population comprises primarily Vδ3 T cells whose lineage and rearrangement are still less well-known [162]. During developmental stages, TCR Vδ2 strands often connect to Vγ9 chains to form Vγ9Vδ2 T cells, which constitute the major component of the γδ T cell population in human peripheral blood [163]. As usual, Vγ9Vδ2 T cells have the ability to recognize infected or transformed cells enriched for phosphorylated antigens (pAgs) or intermediate metabolites such as isopentenyl pyrophosphate (IPP). Tumoral pAgs bind the intracellular domain of BTN3A1, which is associated and comprises complexes with BTN2A1, and BTN2A1 can be recognized directly by TCR Vγ9 chain [164], while IPP is overproduced by the dysregulated mevalonate pathway in cancer cells, which activates Vγ9Vδ2 T cells [165,166]. Instead of pAg-mediated BTN recognition by Vγ9Vδ2 T cells, non-Vγ9Vδ2 T cells with more diverse TCRs usually recognize other molecules like CD1, a family of surface glycoproteins that present with lipid antigens [167]. For example, it has been shown that Vδ1 and Vδ3 subsets recognize CD1 proteins [167,168,169,170,171]. Additionally, BTN molecules were found to be produced within hematological and solid tumors, while CD1 mainly serves as a molecular marker for tumor cells derived from the myelomonocytic and B-cell lineage and is rarely expressed within solid tumors [172,173]. In addition to TCRs, other surface molecules are also expressed by γδ T cells with the purpose of providing additional stimulatory signals and play a role in tumor cells’ recognition as well. Those molecules are usually Natural Killer receptors, such as NKG2D, DNAX Accessory Molecule-1 (DNAM-1), NKp30, NKp44 and NKp46. The ligands for these receptors are often overexpressed by malignantly transformed cells, which are thereby targeted by them and activate them accordingly [174,175,176,177,178,179].

In general, γδ-T cells can use multiple pathways to exert anti-tumor cytotoxic effects. First, activation of the receptor induces perforin expression and release of granzyme B, thereby mediating pore formation at the tumor cell membrane and leading to the entry of granzyme protease into the cell, which subsequently initiates apoptosis of tumor cells [180]. Secondly, TNF-related apoptosis-inducing ligand (TRAIL) and Fas ligand (FasL) expressed by γδ T cells recognize and bind their respective receptors expressed on tumor cells, thus resulting in the induction of apoptosis [180,181,182]. Thirdly, tumors can be targeted by γδ T cells via ADCC. For example, CD16 (Fc receptor III) is expressed on γδ T cells and binds the Fc region of antibodies that is bound to the relevant antigen of the tumor cell, subsequently leading to tumor lysis [180,183,184]. Another co-stimulator is a Toll-like receptor (TLR) that can activate γδ T cells either directly or by participating in the activation of DCs [185]. TLRs feature their structure with integral membrane glycoproteins and thereby are capable of recognizing prototype patterns [185,186]. In response to stimulants such as specific lipid, peptide or nucleic acid PAMPs (pathogen-associated molecular patterns) and DAMPs (danger-associated molecular patterns from damaged tissues), TLRs dimerize and then trigger subsequent signal transduction pathways that lead to the upregulation of inflammatory factors’ expression, like TNF-α, IL-6, IL-1β, IL-12 and IFN-γ [185,186]. However, the consequence of TLR signaling depends on the specific TLR and surrounding circumstances, which can lead to the activation of the immune system or else the suppression.

Cancer cells constitute the main components of the tumor microenvironment, which is surrounded by stromal cells or tissues such as fibroblasts and blood vessels. The microenvironment is usually infiltrated by immune cells and characterized by hypoxia and immunosuppression [187]. Compared with other immune cells that can be hindered by those unfavorable conditions, γδ T cells seem to be more adaptable to this harsh growth environment with hypoxia and inhibition, and still can efficiently identify relevant markers expressed in tumor cells, complete MHC-independent activation and express inflammatory cytokines [188]. It was also shown that activated γδ T cells can stimulate other immune cells and coordinate an antitumor response with them accordingly. For example, M1 and M2 macrophages that infiltrate into the tumor microenvironment could be targeted by γδ T cells and encounter microenvironmental-specific chemokines. Those molecules are able to polarize macrophages towards a pro-tumor M2 state and repolarize them to an anti-tumor M1 state [189]. Additionally, γδ T cells have sorts of immunomodulatory interplay with other immune cells. Vγ9Vδ2 T cells act as APCs, functioning after being activated in a manner similar to DCs, including the increased expression of MHC-I and MHC-II molecules and the release of APC-associated adhesive and costimulatory molecules [190,191]. They may promote their capacity of antigen presentation from both apoptotic and live cancer cells through expressing scavenger receptor CD36 [192]. Then, similar to DCs, the antigen presentation accordingly triggers the differentiation and proliferation of naive CD4^+^ and CD8^+^ αβ T cells [190,191]. Another well-characterized interaction is the crosstalk between γδ T cells and DCs [193], during which Vγ9Vδ2 T cells provide assistance to DC maturation through TNF-α and IFN-γ production, rendering co-stimulatory molecule expression increased in DCs, cytokine secretion enhanced and, as a consequence αβ T cell proliferation and IFNγ production, induced effectively [194]. Through the crosstalk, DCs after maturation activate γδ T cells in turn. γδ T cells also interact with NK cells for their activation. Maniar et al. showed that NK cell cytotoxicity against NK-resistant tumors was increased after NK cells co-cultured with IPP-expanded Vγ9Vδ2 T cells [195]. The crosstalk was performed through co-stimulatory interaction between CD137L on γδ T cells and CD137 on NK cells, which facilitated NK cell activation and the subsequent augmentation of NKG2D expression and increase in antitumor cytotoxicity. An in vivo study as further evidence was reported, showing a key role for γδ T cells in inducing IFNγ production in NK cells during early immune responses [196].

## 6. Eosinophils and Tumor Microenvironment

Eosinophils are a special class of granulocytes that are differentiated in response to allergic reactions, abnormal skin conditions, parasitic and fungal infections, autoimmune diseases, certain cancers and bone marrow diseases. Eosinophils are generally produced in the bone marrow through the differentiation from distinct CD34 myeloid progenitor cells, and normally reside in the connective tissues, specifically in the thymus, gastrointestinal tract, spleen, lymph nodes, ovaries and uterus [197]. Since the late 1900s when eosinophilia was identified in the peripheral blood of cancer patients [198], the tumor-infiltrating eosinophil microenvironment has been shown in sorts of solid tumors, particularly in the tumors of mucosal organs like the gastrointestinal tract and the lungs [199], suggesting that eosinophils are involved in cancer-inflammatory response or are recruited in response to therapies [200].

Clinical and experimental results provided us the potential involvement of eosinophils in the interactions between eosinophils and lymphocyte subsets, including T cells, NK cells and innate lymphoid cells (ILCs) (Figure 5). ILC2s play a role in the mediation of type 2 immune responses and induce chemotactic eotaxin production in epithelial cells and macrophages [201,202] that triggers eosinophil expansion or migration, thereby contributing to the tumor microenvironment in an eosinophil-dependent way or in an eosinophil-independent fashion [203]. The eosinophil–ILC2 interactions were studied mostly in experimental melanoma [204]. For example, one study found that intravenous injection of B16-F10 melanoma cells into mice led to an increase in eosinophilic infiltration into the lungs and upregulated production of IL-5 in lung ILCs [205]. Moreover, in mice lacking IL-5 or treated with anti-IL-5 antibody, the intravenous injection of B16-F10 cells resulted in the decrease in eosinophil infiltration and increase in lung tumor colonization compared with wild-type mice [205]. This suggests that the increased eosinophilic infiltration into the lungs was associated with IL-5 production. In addition, GM-CSF and IL-33 were also shown to be involved in the regulation of ILC2–eosinophil cross-talk in the tumor microenvironment [206]. GM-CSF-deficiency increased tumor growth in tumor-loading mice, and the contribution of GM-CSF production by ILC2s was concerned with the increase in eosinophils and the decrease in tumor growth in tumor-loading GM-CSF-deficient mice [204]. In a lung metastatic murine model by intravenous injections of B16-F10 melanoma cells or Lewis lung carcinoma, ILC2s, eosinophils, CD8^+^ T cells and NK cells were activated and tumor-infiltrating by IL-33 overexpression or IL-33 treatment [207,208].

Eosinophils can interact with NK cells as well through expressing multiple receptors and ligands, such as a CD2 subfamily of receptors, LFA-1. The receptors are involved in the mediation of eosinophil–NK cell cross-talk, while the interruption of eosinophil ligands activating natural cytotoxicity receptors NKp30 and NKp46 in vitro inhibited eosinophil-mediated NK cell activation [209,210,211]. Intravenous injection of 4T1 tumor cells rendered NK cells recruited to the lungs through IL-33 stimulation, which depended on C-C motif chemokine ligand 5 (CCL5) expressed by eosinophils and CD8^+^ T cells. NK cell recruitment was reversed after deleting eosinophils and CD8^+^ T cells [212]. A tumor-cytotoxic effect against K562 myelogenous leukemia cells was shown after co-culturing IL-12-stimulated NK cells with eosinophils, and this cytotoxicity depended on eosinophils’ contact [211]. However, eosinophils may play a suppressive role in NK cell cytotoxicity under the preexistence of type 2 immune responses, which promote tumor growth as a consequence. ILC2-derived IL-5 induced an eosinophil-mediated suppressive effect on lung NK cell activity through modulating the metabolic environment, while the suppressive effect on IFN-γ and granzyme B production by NK cells was attenuated via anti-IL-5 treatment [213].

Bioinformatics data indicated that eosinophils may interact with T cells in cancer. In various primary tumors, through the prediction of the corresponding RNA signatures of multiple immune cells by bulk RNA sequencing and an RNA profiling algorithm, eosinophils, the abundance of such immune cells correlated with the abundance of resting memory CD4^+^ T cells in the primary tumor [199]. Moreover, eosinophil RNA signature may also have correlation with the RNA signatures of CD8^+^ T cells according to bulk RNA-sequence data from pleural metastatic samples of breast cancer patients [214]. A transcriptional profile characterized by M1 macrophages was acquired in IFN-γ-stimulated eosinophils, which presented increased cytotoxicity against colorectal cancer [215,216]. Chemokine expression, like CCL5, C-X-C motif ligand 9 (CXCL9), CXCL10 and CXCL16, were augmented in eosinophils after IFN-γ stimulation, and those chemokines were capable of recruiting T cells [214]. Accordingly, the adoptive transfer of IFN-γ-activated eosinophils into the lungs of tumor-loading mice promoted the infiltration of CD4^+^ and CD8^+^ T cells [214], while pharmacological depletion of regulatory T cells in a tumor model induced evident eosinophilia, which facilitated antitumor immunity [217]. This could demonstrate an anti-tumorigenic function of eosinophil–T cell interactions in the tumor microenvironment. In addition, eosinophils are capable of mediating antigen presentation and, thus, facilitate T cell responses [218].

Clinical studies from independent centers reported that peripheral eosinophil number is upregulated after ICI, especially in the cancer therapies with anti-CTLA4 and anti-PD-1 antibodies [219,220,221,222]. Eosinophils were thereby suggested as cellular biomarkers and end-stage effector cells for positive prognosis in several tumors [223,224,225,226,227,228]. Moreover, human pluripotent stem cell-derived eosinophils presented synergistic tumor-killing capacity with CAR T cells in solid tumors. In the studies with the established tumor-bearing mice loaded with HCT116 (human colorectal carcinoma), MDA-MB-231 (human breast adenocarcinoma) or HepG2 (human hepato-cellular carcinoma), combined therapies with CAR T cells and human embryonic stem cell-derived eosinophils demonstrated superior anticancer efficacy on cancer therapy compared with the single treatment of those alone [229].

## 7. Microbiota and Anti-Cancer Immunity

The parts of the human body, including the digestive system, respiratory system, immune system and female reproductive system, are more or less characterized with colonization by sorts of microorganisms in their mucosal environment. Most studies focusing on the microbiota of the gastrointestinal system indicate that the gut microbiome is actively involved in several host functions, including nutrient absorption, metabolism, immunity and cancer immunology [230].

In the gut microbiota, bacteria play a role in multiple inflammatory diseases of the gastrointestinal tract, and also in the tumorigenesis of the digestive tract. For example, a pronounced correlation has been suggested between *Helicobacter pylori* infection and the development of atrophic gastritis, metaplasia, dysplasia and gastric cancer progression [231,232]. Fusobacterium nucleatum has also been shown to be correlated with tumorigenesis in the gut, extensively in colorectal cancer (CRC). An increased abundance of *F nucleatum* was found in CRC tumors compared with normal tissue. Moreover, enrichment of this species in tumor tissue was involved in the advanced progression from adenomas to carcinomas [233].

In addition, the gut microbiota is also capable of functioning in distance on tumors that are developed in tissues distant from the digestive tract. It has been suggested that *Gammaproteobacteria* can move from the gut to pancreatic tumors, possibly through the duodenum and pancreatic duct, and was involved in efficacy attenuation of the drug gemcitabine there via metabolizing its active form [234]. Tumor growth was promoted by the treatment with commensal microbiota in a Kras transgenic mouse model of PDAC [235]. In this model, on the one hand, the expression of Kras in the mouse pancreas accelerated the malignant development of pancreatic tumors, and on the other hand, caused species changes in the digestive tract microbiome as compared with littermates of wild-type mice, indicating that the gut microbiome alteration could likely be relevant to the enrichment in tumor-promoting bacteria. In a clinical PDAC study, it was shown that the microbiome species composition in the tumor microenvironment was significantly different from the adjacent healthy pancreatic tissue through 16S rDNA sequencing; in addition, some of the bacteria detected in the tumor microenvironment were also detected in the patient’s feces. Specifically, the flora density of *Sachharopolyspora*, *Pseudoxanthomonas*, *Streptomyces* and *Bacillus clausii* in the tumor microenvironment was strongly associated with the prolonged survival of patients [236]. Through further in vivo experiments using fecal microbial transplantation (FMT), mice that were transplanted with FMT from long-term survival patients showed augmentation in tumor infiltration and activation of CD8^+^ T cells, upregulations of serum IFN-γ and IL-2 and decreased tumor burden, while mice that were transplanted with FMT from short-term survival patients showed enhancement in regulatory T cells and myeloid-derived suppressor cells in tumor infiltration and promoted tumor growth [236]. This indicates that gut microbiota from PDAC patients contributed to the modulation of tumor microbiome composition, anti-tumor immunity and tumor growth progression in treated mice.

The gut microbiota is able to have a distant effect as well on liver tumorigenesis via the transportation of pathogen metabolites, the associated molecular patterns and antigens from intestinal bacteria through the hepatic portal vein to the liver. It has been shown that secondary bile acids (BAs), such as lithocholic acid or murine-specific ω-muricholic acid, produced by Clostridium species presented pro-tumor effects in sorts of preclinical models of primary or metastatic liver cancer [237]. The mechanism was inferred to be related to the secondary BA-induced decrease in CXCL16 expression in liver sinusoidal endothelial cells, which attenuated recruitment of NK T cells to the liver. In fact, overproduction of the secondary BA deoxycholic acid could be due to the enrichment of *Clostridium* species in the gut caused by a high-fat diet, and deoxycholic acid accumulation resulted in the stimulation of the TLR2 pathway, leading to COX2 overexpression and, thereby, prostaglandin E2 (PGE2) overproduction in consequence [238]. PGE2 is shown to play a role in accelerating tumor progression through the suppression of tumor immunity [238]. Therefore, accumulation of the secondary BA deoxycholic acid is involved in pro-tumor effects on carcinogen-induced liver cancer.

It has been hypothesized that digestive tract microbiota may influence the biological functions and processes of all systems in the host through different signaling mechanisms. For example, the digestive tract microbiome may be involved in the modulation of sex hormone level through metabolizing and synthesizing and may signal to the central nervous system and potentially facilitate the progression of hormone-related cancers, including endometrial, ovarian, prostate and breast cancer [239,240,241]. It has been also shown that bacterial metabolites such as short-chain fatty acids, secondary Bas and enterolignans could influence tumor development and progression through the regulation of immune-mediated signals via host circulation access [242,243]. The gut microbiota is therefore regarded to constitute a reciprocal circuitry called the gut–brain axis [244,245]. Neuroinflammation induced by the gut microbiota could be contributing to the pathogenesis of diverse central nervous system diseases. An in vivo study showed that inhibition of the immunosuppressive effect of myeloid-derived suppressor cells caused the triggering of activation of the reward system of the brain in mice, which resulted in the alleviation of lung tumor growth [246]. This may indicate that the gut–brain axis could be a pathway via which the gut microbiota could influence the reward system through arousing immunosuppressive effects and contribute to cancer progression systemically [247]. Tumor development is intricately relevant to the host immune system, which is intimately influenced by the gut microbiota, leading to the potential involvement of the gut microbiota in tumor progression and therapeutic efficacy through direct intervening by bacterial metabolites or indirect alteration of immune homeostasis (Figure 6). This is evidently supported by preclinical and clinical studies in various cancers [234,235,248,249,250,251,252,253].

## 8. Conclusions

A good understanding of the mechanisms of cancer immunity could provide therapeutic opportunities. In general, the strategy includes shutting immune suppression and restoring antitumor immune responses through modulating the activities of the components involved in the immune landscape of cancer.

The synergistic action of both arms, innate and adaptive immunity, forms the basis of the carefully arranged anti-tumor response of the host immune system. The CTLs select for target killing by recognizing the tumor antigen peptides presented by MHC-I, subsequently killing tumor cells and providing support for anti-tumor immunity. MHC-I deficiency in cancers is commonly caused mostly by NLRC5 loss, which has been regarded as poor prognosis for many cancers. Restoring NLRC5 expression can be helpful for tumor immunogenicity and the recognition of cancer antigenic peptides. Although MHC-II, canonically expressed by APCs, has been shown to play multiple key roles in antitumor immunity, tsMHC-II, expressed on various tumor cells, exhibits a better prognostic correlation with cancer patients, both with patient survival and with response to immunotherapy with anti-PD-1/anti-PD-L1 drugs. TsMHC-II upregulation can augment cancer immunity and, thus, could be a novel strategy to ameliorate anti-tumor immune responses. TsMHC-II holds promise as a predictive biomarker of cancer immunotherapy compared with other parameters. NK cells are considered as crucial members of the complex immune response: in the primary immune response, via directly making lysis of cancer cells through the ADCC pathway and removing MHC-damaged cells; for the secondary adaptive immune response, by priming the tumor microenvironment via inducing PD-L1 expression on tumors and the recruitment of DCs and T cells. Therefore, without an NK cell compartment, it could be able to neither mount killing of malignant cells nor harness the full efficacy of immunotherapies. It is thus necessary to reconstitute the NK cell compartment via precursory adoptive NK cell transfer, which triggers the priming of the immune system for optimal effector populations’ activity and therapeutic efficacy. The differentiated γδ T cell population initially produces antitumor effects, which, however, can be restricted by their low proliferation and self-renewal capability, rendering their effect less persistent. In contrast, γδ T cells in naive or central memory status have long persistence and are amenable to continuous tumor surveillance. After antigen stimulation, this cell population can differentiate into effector cells that recognize tumors and then retain a subpopulation of self-renewing memory cells for subsequent tumor encounters and corresponding responses. Therapeutically, naïve γδ T cells or those in central memory status are applicable for clinical use, but generating a large enough population in vitro for therapeutic desire could still be challenging. It is suggested by the available data that eosinophils are involved in the facilitation of tumor immunity via cross-talk with various lymphocyte subsets directly and indirectly. But, the transcriptomic profile and individual heterogeneity of eosinophils are far from being elucidated, for eosinophils are often undetectable by most single-cell RNA-sequencing analyses [254], rendering us limited to fully elucidate unique eosinophil subsets, eosinophils interacting with specific subsets in the tumor microenvironment or their distinct functions. New experimental approaches are required to overcome these challenges. Commensal microbiota, via their pronounced impact on host immunity, have been shown to influence anti-tumor immunity and therapeutic efficacy. Clinically, they are mainly based on the application of microbial agents with immunostimulatory effects to enhance the efficacy of cancer immunotherapy. Further understanding microbe-mediated immunoregulatory pathways and components, and discovering strains with precise immunostimulatory and immunosuppressive effects, may contribute to facilitating precise and personal therapeutic approaches that may render fewer undesirable outcomes. In addition, it is plausible to inhibit the proliferation and decrease the abundance of immune-suppressing or tumor-promoting bacteria, which may enhance immunotherapeutic efficacy.

## Figures and Tables

**Figure 1 cells-12-01882-f001:**
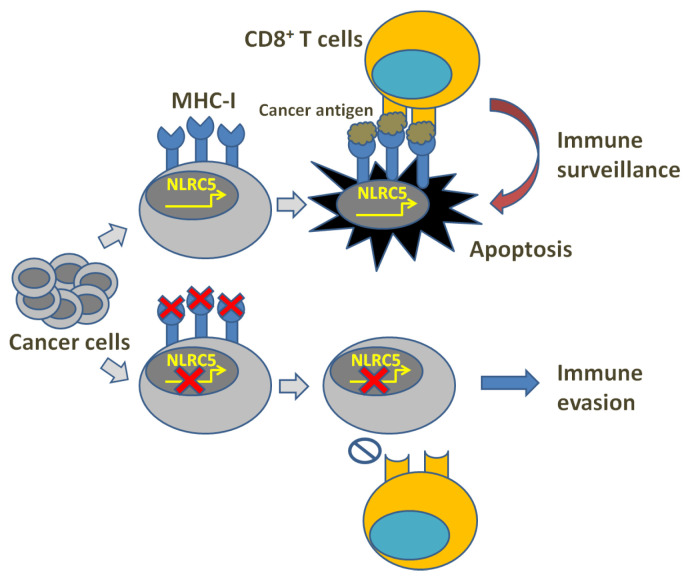
Illustration of MHC-I deficiency and immune evasion.

**Figure 2 cells-12-01882-f002:**
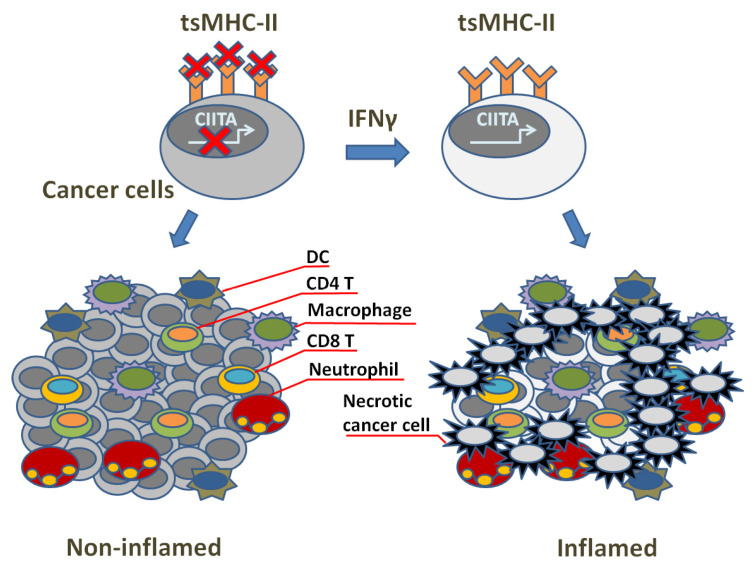
Illustration of tsMHC-II’s role in tumor immune surveillance.

**Figure 3 cells-12-01882-f003:**
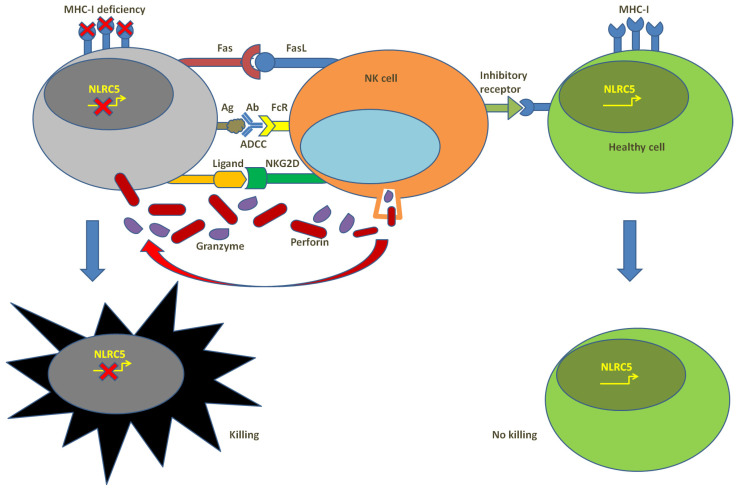
Summary of NK cells’ function in anti-cancer immunity.

**Figure 4 cells-12-01882-f004:**
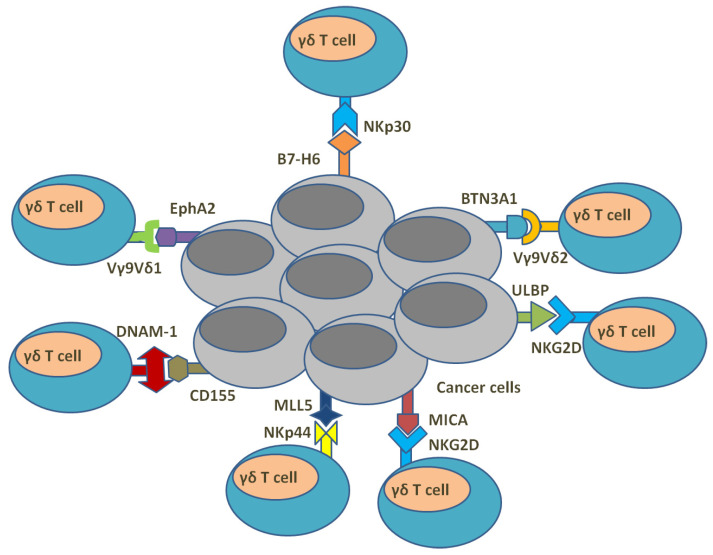
Summary of the interaction between γδ T cells and cancer cells.

**Figure 5 cells-12-01882-f005:**
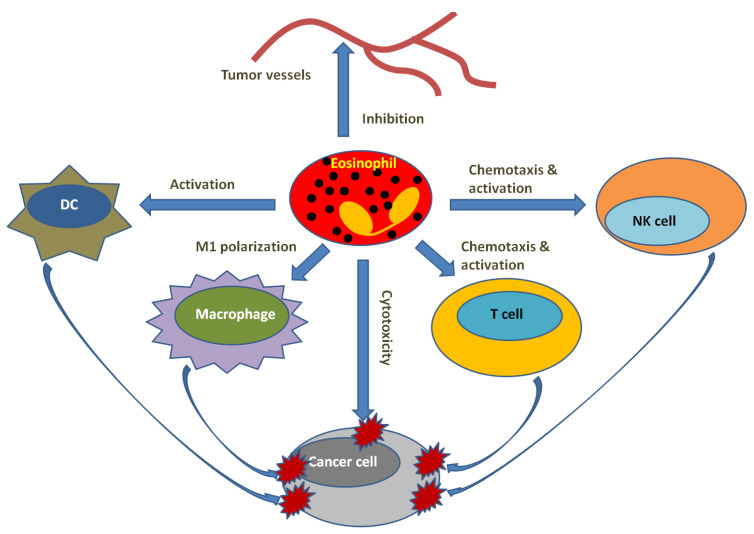
Illustration of eosinophils’ role in tumor microenvironment.

**Figure 6 cells-12-01882-f006:**
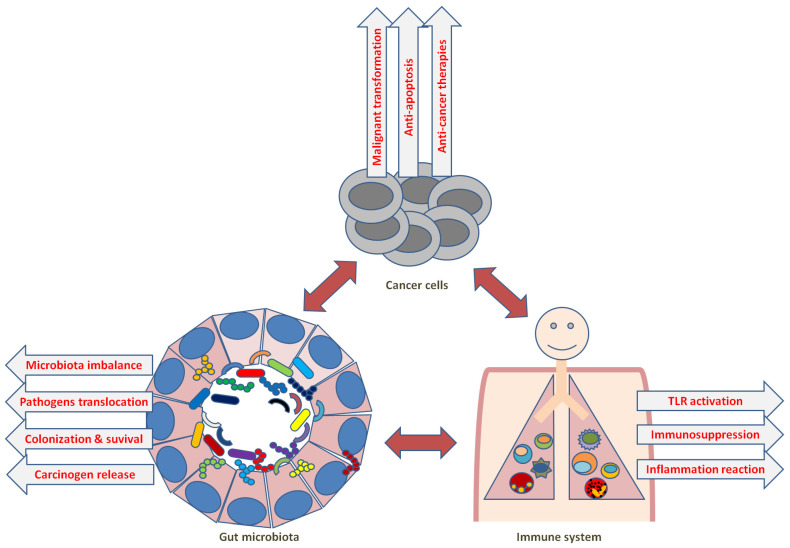
Summary of the influence of commensal microbiota in cancer immunity.

## Data Availability

Not applicable.

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
