# Peer review of "Insight into Cancer Immunity: MHCs, Immune Cells and Commensal Microbiota"

_cells, 2023, doi:10.3390/cells12141882_

Round 1

Reviewer 1 Report

Wen et al provide a comprehensive review of the literature concerning cancer immunity. Clearly a substantial amount of effort has been expended on this. Unfortunately, the review is very hard to read, there are an abundance of subjects covered, and a substantial number of points are made, however, there is too little explanation provided - it comes across as a succession of facts. Without expanding upon key points with greater explanation, I suspect readers will become lost.

Perhaps it might be better for the authors to consider including fewer sections in their review e.g. maybe MHC-I and MHC-II, and the other sections might be discussed in a separate review (NK, gamma delta T, eosinophils, microbiota)? This might allow an opportunity to expand upon their key points in greater detail. Perhaps even including some example figures? For example, a figure describing the NLRC5 regulation of MHC-I might be advantageous.

There are a number of incorrect or misleading statements that require correcting, but I have not listed these here, as there are more substantial changes to be made. There is also quite a lot of abbreviations used without explanation upon the first use. This should be corrected. I am happy to provide these if required.

Generally, the quality of the English language was very good. There were a number of corrections that might be made, but this is not my main concern. Instead, I think the biggest problem is too much is being covered, which does not allow enough scope to discuss subjects appropriately.

Author Response

Comments and Suggestions for Authors
Wen et al provide a comprehensive review of the literature concerning cancer immunity. Clearly a substantial amount of effort has been expended on this. Unfortunately, the review is very hard to read, there are an abundance of subjects covered, and a substantial number of points are made, however, there is too little explanation provided - it comes across as a succession of facts. Without expanding upon key points with greater explanation, I suspect readers will become lost.

Perhaps it might be better for the authors to consider including fewer sections in their review e.g. maybe MHC-I and MHC-II, and the other sections might be discussed in a separate review (NK, gamma delta T, eosinophils, microbiota)? This might allow an opportunity to expand upon their key points in greater detail. Perhaps even including some example figures? For example, a figure describing the NLRC5 regulation of MHC-I might be advantageous.

There are a number of incorrect or misleading statements that require correcting, but I have not listed these here, as there are more substantial changes to be made. There is also quite a lot of abbreviations used without explanation upon the first use. This should be corrected. I am happy to provide these if required.

Re: Thanks for the comments and suggestions provided. In fact, I agree with you that the review focusing on fewer sections can be allowed an opportunity to expand more key points and in greater details, which is suitable for the readers who have deeply experienced in those research topics that are covered by the review. Nevertheless, for the readers who have little experienced in cancer immunity, our current manuscript as a review could be more readable, easier to understand, and more comprehensive in study topic in cancer immunity area.
Besides, manuscript focusing on few subject as a separate review, could be a good trying later, but not at present. Otherwise, this revision can be almost regarded as a rewritting.
After all, you gave good suggestions and worthy of consideration.
The explanation for the abbreviations has been complemented.
Thanks very much.

Comments on the Quality of English Language
Generally, the quality of the English language was very good. There were a number of corrections that might be made, but this is not my main concern. Instead, I think the biggest problem is too much is being covered, which does not allow enough scope to discuss subjects appropriately.

Re: Thanks for your comments. 

Reviewer 2 Report

This article provides a review of relevant literature on the subject of immune surveillance in cancer and the tumor microenvironment. The potential role of commensal bacteria in the gut microbiome in discussed. The review is focused, thorough, well organized, and should be of interest to readers. There are a few minor editorial issues to clean up.

Line 15 – accessorily? intricating?

Line 79 – cisregulatory? cis-regulatory

Line 85 – spelling – wildtype

Line 102 – meditation?

Line 505 – spelling – accelerated

Some additional editing would be useful, especially use of singular versus plural nouns, use of "the" to mark proper nouns, and reduction of some on the lengthy sentences. These changes would make the article more easily readable.

Author Response

This article provides a review of relevant literature on the subject of immune surveillance in cancer and the tumor microenvironment. The potential role of commensal bacteria in the gut microbiome in discussed. The review is focused, thorough, well organized, and should be of interest to readers. There are a few minor editorial issues to clean up.

Re: Thanks for the comments and suggestions. The minor issues have been corrected as suggested.

Reviewer 3 Report

The authors summarize the role of the immune system, tumor microenvironment and microbiota and their potential therapeutic impact in cancer patients. This review article allows for an overview about the diffrent targets of immunotherapy and the underlying biologic mechanisms. To my opinion the manuscript is siutable for publication in Cells.

Author Response

Comments and Suggestions for Authors
The authors summarize the role of the immune system, tumor microenvironment and microbiota and their potential therapeutic impact in cancer patients. This review article allows for an overview about the diffrent targets of immunotherapy and the underlying biologic mechanisms. To my opinion the manuscript is siutable for publication in Cells.

Re: Thanks very much for your comments.

Round 2

Reviewer 1 Report

The manuscript has been revised. In the first review, I had suggested that the review might be easier for readers if it contained fewer subjects. The authors have chosen not to take this suggestion. I fear readers may find the text extremely difficult to read and follow. The authors have chosen to include six figures in the revised manuscript. Unfortunately, these are not included in the manuscript, and it is not clear how to access them. I had emailed the editorial office to ask how to access the figures, but got a response asking me to submit my completed review. I am unable to endorse this for publication without seeing the new figures.

Minor improvements might be made in places, but generally good.